# Unraveling Spontaneous Coronary Artery Dissection in Sudden Cardiac Death: Integrating Pathology, Genetics, and Molecular Autopsy

**DOI:** 10.3390/ijms262211072

**Published:** 2025-11-16

**Authors:** Cecilia Salzillo, Andrea Quaranta, Gerardo Cazzato, Andrea Marzullo

**Affiliations:** 1Department of Experimental Medicine, Ph.D. Course in Public Health, University of Campania “Luigi Vanvitelli”, 80138 Naples, Italy; cecilia.salzillo@unicampania.it; 2Department of Precision and Regenerative Medicine and Ionian Area, Pathology Unit, University of Bari “Aldo Moro”, 70124 Bari, Italy; a.quaranta35@studenti.uniba.it (A.Q.); gerardo.cazzato@uniba.it (G.C.)

**Keywords:** spontaneous coronary artery dissection, sudden cardiac death, coronary vessels, pathology, genetics, molecular autopsy

## Abstract

Spontaneous coronary artery dissection (SCAD) is a non-atherosclerotic cause of acute coronary syndrome, characterized by the development of a false lumen within the coronary arterial wall, leading to narrowing or complete occlusion of the true lumen. This underrecognized condition accounts for a substantial proportion of sudden cardiac death (SCD), particularly among young, otherwise healthy women. Macroscopically, SCAD is defined by intramural hematoma and focal thickening of the arterial wall, while histological examination demonstrates separation of the tunica media, elastic fiber degeneration, and variable inflammatory infiltrates. Proposed pathogenic mechanisms include primary intimal tear and primary intramural hematoma, frequently associated with predisposing conditions such as fibromuscular dysplasia, connective tissue disorders, and specific hormonal states. In cases of myocardial infarction, the myocardium exhibits acute ischemic necrosis and early hypoperfusion injury. Postmortem diagnosis requires meticulous coronary dissection, adjunctive histochemical and immunohistochemical staining, and, when indicated, molecular autopsy (MA). The purpose of this review is to provide an updated synthesis of current knowledge on SCAD as a cause of SCD, integrating pathogenetic, morphological, and genetic perspectives, and to emphasize the role of MA as both a diagnostic and preventive tool.

## 1. Introduction

Spontaneous coronary artery dissection (SCAD) is a rare but potentially life-threatening cardiovascular emergency, characterized by the creation of a false lumen within the coronary arterial wall. Expansion of this false lumen may compress the true lumen, thereby reducing or obstructing myocardial blood flow and leading to ischemia, acute myocardial infarction, and, in the most severe cases, sudden cardiac death (SCD) [1,2].

SCAD represents a distinct clinical entity, separate from atherosclerotic coronary artery disease, with a multifactorial etiology arising from the interplay of triggering events, predisposing conditions, and intrinsic structural alterations of the arterial wall [3,4]. The condition predominantly affects women, with a peak incidence during childbearing years and the peripartum period. In these settings, hormonal and hemodynamic changes, including increased plasma volume, alterations in vasomotor tone, and variations in coronary flow, may transiently weaken the arterial wall. Approximately 70% of pregnancy-related cases occur in the early postpartum period, even in the absence of obstetric complications [5,6].

Among the conditions most frequently associated with SCAD, fibromuscular dysplasia (FMD) is reported in up to 86% of affected patients and is considered an independent predictor of major adverse cardiovascular events [7,8]. Less frequently, autoimmune and inflammatory disorders such as systemic lupus erythematosus and vasculitis, as well as inherited connective tissue diseases—including vascular Ehlers–Danlos syndrome, Marfan syndrome, and Loeys–Dietz syndrome—have been implicated as predisposing factors [9,10]. Although the majority of SCAD cases are sporadic, pathogenic genetic variants may account for up to 17% of cases, supporting the role of genetic testing in clinical management and in the prevention of recurrent cardiovascular events [5,6].

The aim of this review is to provide an updated and critical synthesis of the current knowledge on SCAD as a cause of SCD, with particular emphasis on anatomo-pathological features, pathogenetic mechanisms, and emerging genetic evidence. Furthermore, we underscore the increasing role of molecular autopsy (MA) as a complementary tool to conventional autopsy, capable of enhancing diagnostic accuracy, supporting family screening strategies, and opening new perspectives for prevention. By integrating morphological and genetic insights, this review seeks to offer a comprehensive framework for both pathologists and clinicians engaged in the management of SCAD and its potentially fatal outcomes. Table 1 summarizes the key differences between SCAD and coronary atherosclerosis.

## 2. Materials and Methods

This narrative review was conducted through a structured literature search aimed at identifying studies addressing SCAD, its pathological and genetic mechanisms, and the role of MA.

PubMed/MEDLINE, Scopus, and Web of Science databases were searched using combinations of the following keywords and MeSH terms: “spontaneous coronary artery dissection”, “SCAD”, “sudden cardiac death”, “molecular autopsy”, “fibromuscular dysplasia”, “connective tissue diseases”, “genetic testing”, and “forensic pathology”. Boolean operators (“AND”, “OR”) were used to combine search terms.

Inclusion criteria included original research articles, case series, reviews, and official statements in English that discussed the epidemiology, pathology, genetics, or postmortem diagnosis of SCAD. Exclusion criteria included studies for which full-text access was not available, articles not written in English, non-peer-reviewed articles, and publications not relevant to the topic.

The synthesis followed a thematic framework, organized into sections that reflected the main aspects of SCAD: pathogenesis, pathological features, postmortem diagnosis, molecular autopsy, and forensic implications.

## 3. Underlying Pathogenetic Pathways

SCAD results from a complex, multifactorial pathological process in which structural vulnerability of the arterial wall interacts with systemic predisposing factors and acute triggers. Two main pathogenetic mechanisms have been recognized: (1) primary intimal rupture, which permits direct blood entry from the true lumen into the coronary wall, creating a false lumen, and (2) primary intramural hematoma, caused by bleeding of the vasa vasorum in the absence of intimal disruption [11,12].

Structural predisposing factors include congenital or acquired alterations of the vascular extracellular matrix. FMD, for example, produces multifocal or focal lesions in medium and small-sized arteries, reducing their resistance to hemodynamic stress [13,14]. Inherited connective tissue disorders are also implicated, including vascular Ehlers–Danlos syndrome caused by *COL3A1* mutations, Marfan syndrome due to *FBN1* mutations, and Loeys–Dietz syndrome associated with mutations in genes of the TGF-β pathway. These conditions result in disorganization of medial fibrillar architecture and reduced mechanical strength, predisposing the arterial wall to dissection [15,16].

Hormonal influences play a significant role. Pregnancy and the postpartum period are associated with increased plasma volume, alterations in vasomotor tone, and transient vascular wall remodeling. Combined with the hemodynamic stress of delivery, these changes create a temporary vulnerability of the coronary arteries [17].

Acute triggers include abrupt increases in intraluminal pressure and shear stress, as observed during strenuous physical exertion, emotional stress, Valsalva maneuvers, or childbirth. In patients with underlying arterial fragility, such triggers may precipitate vascular layer separation [1,12].

The genetic architecture of SCAD is complex, involving rare high-impact variants, lower-penetrance variants, and polygenic contributions. A systematic review published in 2023 reported that, among SCAD patients who underwent genetic testing, 20% harbored mutations in *COL* genes, 13.7% in *TLN1*, and 8.4% in *TSR1* [18]. These findings indicate that, although highly penetrant mutations are present, they account for only a minority of cases. In parallel, a genome-wide association study (GWAS) conducted in 2023 estimated a polygenic heritability of ~71% for SCAD, identifying at least 16 risk loci that collectively explain about one-quarter of its genetic heritability [19,20].

Genetic correlations have also been demonstrated between SCAD and other vasculopathies, including arterial dissections, FMD, stroke, and intracranial aneurysms, suggesting shared biological and structural mechanisms [19]. Notably, SCAD differs from atherosclerotic coronary artery disease (CAD): some SCAD-associated loci overlap with CAD loci but carry opposite risk alleles, resulting in an overall negative genetic correlation between the two conditions [19]. Among common genetic risk factors, hypertension has emerged as a plausible contributor, since variants associated with elevated blood pressure increase SCAD risk, whereas traditional cardiovascular risk markers such as LDL cholesterol, obesity, diabetes, and smoking do not appear to exert a significant genetic influence [19].

Additional studies in smaller cohorts, mainly in Italian populations, have employed whole-exome sequencing (WES) and TRIO-WES approaches. These investigations identified potentially pathogenic variants in genes linked to connective tissue disorders (*COL3A1*, *COL1A2*, and *SMAD3*), as well as variants of uncertain significance in other genes associated with SCAD and vascular fragility. Interestingly, premutations in the *FMR1* gene have also been detected, raising the possibility of underexplored mechanisms such as microRNA dysregulation or RNA-mediated toxicity from repeat expansions [20].

Table 2 summarizes the major genes associated with SCAD.

## 4. Pathological Features

Macroscopically, SCAD is characterized by an intramural hematoma located most often between the media and adventitia, and more rarely between the intima and media. The accumulation of blood produces segmental thickening of the arterial wall and concentric or eccentric narrowing of the true lumen. The length of the dissection may range from a few millimeters to several centimeters, occasionally extending to multiple vessels. The left anterior descending artery is the most frequently involved, reported in 32–46% of cases [21].

On cross-section, the false lumen appears as an irregular red-purple or brown cavity compressing the true lumen. When an intimal rupture is present, a communicating orifice between the true and false lumen can be identified. In cases of primary intramural hematoma, the endoluminal surface remains intact, displaying a smooth, elongated stenosis.

Histologically, SCAD is defined by separation of the elastic and collagen fibers within the media, with interposition of clotted blood or serous fluid. The internal elastic lamina may appear disrupted or distorted. Focal myxoid degeneration of the media and fragmentation of elastic fibers are frequent findings [15]. The inflammatory infiltrate, when present, is typically mild and composed of T lymphocytes, macrophages, and occasional eosinophils. The latter has been interpreted as either an early feature or a marker of vascular remodeling. In some cases, structural changes consistent with fibromuscular dysplasia are observed, including medial disorganization, fibrous thickening, and rarefaction of smooth muscle cells [15].

In cases associated with connective tissue disorders, the elastic and collagen fibers exhibit reduced fibrillar density and altered architecture, reflecting congenital fragility of the arterial wall [15]. In fatal cases complicated by sudden cardiac death, coronary findings are accompanied by acute myocardial ischemic injury characterized by transmural or subendocardial coagulative necrosis, waviness of myocardial fibers, interstitial edema, and capillary congestion. In very rapid deaths, only early alterations may be detected, such as loss of cross-striations and a modest neutrophilic infiltrate [21,22].

Table 3 summarizes the main features of SCAD.

## 5. Postmortem Diagnosis

Recognition of SCAD at autopsy represents a considerable diagnostic challenge, particularly in cases of SCD in young individuals without significant atherosclerosis. The initial step is macroscopic examination of the coronary arteries through serial dissection, sectioning the vessels at close intervals (generally 3–5 mm) along the entire course of the main arteries and their branches. This approach allows identification of intramural hematoma or a false lumen, typically appearing as an irregular, brownish to purplish thickening of the arterial wall. When dissection originates from an intimal tear, a small fissure may be observed connecting the true lumen with the false lumen [22].

Once a suspicious area is detected, targeted sampling is performed for histological evaluation. Hematoxylin and eosin (HE) staining provides an initial overview of the vascular morphology, highlighting separation of the tunica media, elastic fiber fragmentation, and any inflammatory infiltrates. To achieve greater structural resolution, special stains are employed: Elastic Van Gieson (EVG) to visualize elastic fibers and detect breaks or deformations of the internal elastic lamina, and Masson’s trichrome (MT) to assess the distribution and density of collagen fibers [15,21].

Immunohistochemistry can further refine the diagnosis. Markers such as CD3 and CD68 help characterize inflammatory infiltrates, distinguishing lymphocytic from macrophage components, while anti-CD31 and anti–von Willebrand factor (VWF) antibodies provide information on endothelial integrity and neovascularization. Examination of the vasa vasorum may be informative when intramural bleeding is suspected as the initiating event [21,22].

In recent years, advanced postmortem imaging techniques have expanded the diagnostic armamentarium. Postmortem micro-CT enables high-resolution three-dimensional reconstructions of the coronary wall without compromising tissue integrity, whereas autopsy coronary angiography with contrast medium delineates the distribution of the false lumen and associated structural alterations, facilitating correlation between anatomical and pathogenetic findings [23].

When macroscopic and microscopic findings remain inconclusive, MA offers an invaluable tool for detecting pathogenic variants. Beyond clarifying the cause of death, MA has important implications for surviving family members, enabling targeted screening programs and transforming forensic investigation into a preventive strategy for public health [24].

In summary, postmortem diagnosis of SCAD requires an integrated approach combining meticulous gross examination with histological, immunohistochemical, advanced imaging, and genetic analyses. This multidimensional strategy enhances the contribution of pathology not only to disease understanding but also to prevention of recurrent events in at-risk families. Figure 1 illustrates the key steps of the integrated postmortem diagnostic pathway.

## 6. Molecular Autopsy as a Diagnostic and Preventive Tool in Sudden Cardiac Death

MA, based on post-mortem genetic testing through next-generation sequencing (NGS) or WES, is emerging as an essential tool in the investigation of SCD, particularly when conventional autopsy is negative, as is often the case in younger individuals or in structurally normal hearts. By identifying pathogenic variants, MA can clarify the underlying etiology of sudden death and provide opportunities for cascade genetic screening in relatives, enabling targeted preventive interventions [25,26,27,28,29].

In addition to the classical postmortem genetic testing focused on the main channelopathy genes (*KCNQ1*, *KCNH2*, *SCN5A*, and *RYR2*), the progressive implementation of next-generation sequencing (NGS) technologies has expanded the scope of molecular autopsy. This approach now includes the investigation of genes involved in connective tissue integrity and vascular fragility, such as *COL3A1*, *FBN1*, *TLN1*, *SMAD3*, and *FMR1*, which are particularly relevant to the pathogenesis of SCAD. These technological advances have significantly increased the diagnostic yield and preventive potential of molecular autopsy.

A recent multicenter study showed that MA can detect pathogenic variants in approximately 20% of cases without evident structural abnormalities, confirming its diagnostic and preventive value [30]. In particular, its application in athletes who died suddenly identified clinically relevant variants in 17% of cases, mainly in genes associated with cardiomyopathies rather than ion channelopathies [31]. Similarly, a review highlighted that in young individuals with sudden unexplained death (SUD), up to 30% of cases classified as sudden arrhythmic death syndrome (SADS) could be explained by post-mortem genetic testing, thereby supporting surveillance of family members at risk [32].

Methodologically, conventional targeted gene panels have largely been replaced by NGS, which enables the simultaneous analysis of dozens to hundreds of genes, making the process faster, more cost-effective, and more accurate. However, the interpretation of variants of uncertain significance (VUS) remains a major challenge, requiring evaluation by multidisciplinary teams including geneticists, cardiologists, and pathologists [27,29,33,34,35,36]. One of the most recent studies, conducted in young patients with apparently normal hearts, used WES combined with a customized virtual gene panel, achieving a diagnostic yield of 80% in cases with structurally intact hearts, demonstrating the efficacy of this advanced molecular approach [37].

In forensic medicine, a prospective study of individuals who died suddenly at ≤50 years of age reported positive genetic findings in 7.6% of those aged ≤35 years and in 4.9% of those aged 36–50 years, confirming that MA provides valuable diagnostic insights even in carefully selected cases [38]. Finally, a systematic review has underlined the importance of including MA in standard autopsy protocols, emphasizing its diagnostic, clinical, and preventive benefits for both selected cases and the broader population affected by SCD [39].

SCD represents the most dramatic and feared outcome of SCAD [40,41,42]. Although not all patients with SCAD experience a fatal event, complete obstruction of the coronary lumen may precipitate severe myocardial ischemia, rapidly leading to ventricular fibrillation or asystole. Such events are frequently fatal, particularly when they occur outside hospital settings where the chain of survival cannot be effectively implemented [43,44,45].

From a pathological perspective, SCAD-associated SCD is characterized by clear macroscopic findings: the false lumen is distended and filled with thrombus, with marked compression of the true lumen. In cases of intimal rupture, direct communication between the two lumina may be evident, whereas in cases of primary intramural hematoma the endoluminal surface often appears intact, making diagnosis more challenging and highlighting the need for meticulous coronary examination [46].

Histological evaluation commonly demonstrates acute ischemic myocardial injury. When death occurs a few hours after symptom onset, findings include coagulative necrosis, interstitial edema, and neutrophilic infiltration. In cases of near-instantaneous death, only early changes such as waviness of myocardial fibers, loss of cross-striations, and initial cytoplasmic hypereosinophilia may be observed. Although non-specific, these features indicate early ischemic damage and must be correlated with the location of the coronary lesions [47,48,49].

From a clinical-forensic perspective, establishing SCAD as the cause of death enables exclusion of alternative sudden causes such as primary arrhythmias or coronary embolism, thereby providing a precise pathogenetic explanation. Moreover, identifying predisposing conditions such as FMD [7,22,49,50,51,52], connective tissue disorders [53,54], or vasculitis [55,56] is crucial for initiating family screening programs aimed at prevention.

Experience indicates that SCAD is frequently underdiagnosed in cases of SCD, particularly in young women. The absence of standardized autopsy protocols increases the risk of misclassification, often resulting in the cause of death being recorded as undetermined. The combined application of dedicated autopsy protocols, advanced post-mortem imaging techniques, and MA can substantially reduce this diagnostic gap [22,25,57,58,59].

In conclusion, consideration of SCAD in post-mortem practice is critical: recognizing it as the cause of SCD transforms an unexplained death into an accurate diagnosis, with major implications for prevention and family management. In this context, the pathologist is not merely a lesion describer but an integral contributor to public health strategies aimed at reducing the burden of sudden death in the population.

## 7. Forensic Implications of Postmortem SCAD Diagnosis and Molecular Autopsy Practices

The increasing diagnosis of SCAD is a significant cause of acute coronary syndrome and, in selected cases, SCD, raises complex medicolegal issues that impact both forensic autopsy practice and the protection of surviving family members.

From a medicolegal perspective, accurately identifying SCAD as the cause of death requires dedicated autopsy protocols with serial coronary inspection, close-up sectioning, targeted special stains, and postmortem imaging, as well as organizational and regulatory procedures that ensure accuracy, traceability, and subsequent preventative actions for at-risk family members. Failure to perform adequate coronary investigations or adopt an integrated approach that includes MA may result not only in diagnostic error but also, under certain circumstances, legal consequences for the healthcare institution and the forensic pathologist in charge of the investigation [24,29,38].

First, the role of MA has important implications for both accountability and prevention. Several recent studies demonstrate that MA significantly increases the diagnostic yield in SUD and SCDY, allowing the identification of inherited causes not evident through macroscopic or histological examination alone, such as cardiomyopathies, channelopathies and genetic variants associated with vascular fragility relevant to dissections such as SCAD [27,29,33,38]. The lack of standardization of MA, when clearly indicated by clinical and autopsy circumstances, may expose institutions to legal challenges since the failure to perform genetic tests that could have identified a hereditary condition and triggered cascade screening in relatives could be interpreted as a failure to adopt appropriate measures to protect the health of the family [9,19].

Second, performing MA entails specific procedural and ethical-legal obligations, such as preservation and chain of custody of biological samples, informed consent for sample collection and analysis, and secure management of sensitive genetic data in compliance with privacy regulations and national standards for medical records and archiving. In the legal field, the requirement for judicial authorization may vary; this requires clear policies and shared protocols between pathologists, geneticists, judicial authorities, and prevention services to avoid delays or disputes regarding the legality of postmortem genetic testing [32,33,38]. Complete documentation of each stage, including custody and transfer records, is crucial evidence in legal proceedings.

A further critical issue concerns the interpretation of the results, particularly of VUS. The widespread use of NGS produces a considerable percentage of VUS and their communication to family members and their evidentiary value entail medico-legal risks [33]. Prematurely attributing a VUS as the probable cause of death may lead family members to make inappropriate clinical or psychological decisions, while omitting potentially relevant findings may be considered an omission. To mitigate the risk, interpretation should be performed by multidisciplinary teams including a pathologist, a clinical geneticist, a cardiologist, an ethical–legal consultant, and communication with families should be accompanied by structured genetic counseling and periodic re-evaluation of variants in light of evolving evidence [32].

Further implications concern the duty to inform/the duty to warn family members who are potentially carriers of pathogenic variants. When MA identifies a clearly pathogenic variant associated with inherited cardiac or vascular risk, institutions and professionals are required to activate procedures to communicate the finding, facilitate access to clinical and genetic evaluation, and document the actions taken. Failure to provide such information may result in civil or administrative liability, particularly if the omission has prevented the adoption of effective preventive measures such as screening, follow-up, prophylactic therapies or lifestyle interventions [27,29,32].

Another aspect that is often underestimated is the relationship between pathological findings and clinical responsibility. In cases of SCAD death, the forensic investigation is not only aimed at establishing the mechanism of death, but also at evaluating hospital practices, such as the accuracy of the antemortem diagnosis, the timeliness of treatment, and the appropriateness of interventions. Civil parties may claim negligence or omissions in clinical management prior to death; for this reason, the autopsy report must not only clarify the pathological diagnosis but also evaluate the causal link or its absence between the clinical interventions and the fatal outcome [9].

Ethical and legal considerations are equally critical, as balancing the genetic privacy of the deceased with the health interests of the family requires explicit policies. International guidelines emphasize the importance of proactive family counseling, secure data protection, and defined limits on access to genetic information [28]. Procedures should include the appointment of a communications officer and protocols for accurate and sensitive disclosure.

From a legal perspective, in contexts such as Italy, Law 24/2017 (the so-called Gelli-Bianco Law) and subsequent court rulings have influenced the assessment of medical negligence and due diligence. Failure to comply with recognized guidelines and protocols can be a central element in litigation. Applied to the postmortem investigation, the absence of standardized autopsy protocols for SCD and the failure to integrate MA where indicated may be interpreted as a failure to adhere to professional standards. Therefore, the adoption of national protocols for the investigation of SCD and the systematic use of MA represent not only a clinical recommendation, but also a legal protection for professionals and institutions [19,38,59,60,61].

Finally, the economic and organizational implications must not be overlooked. The systematic implementation of MA and structured family counseling requires financial investment, accredited laboratories, qualified personnel, and referral networks for family screening. While the costs are significant, preventing future events among family members can reduce long-term healthcare burdens and liabilities [9,29,38]. At the public health level, the creation of national SCD registries and the integration of clinicopathological and genetic data are essential for risk surveillance and prevention strategies at the population level [27,31,32,42,61,62,63].

In conclusion, the medicolegal implications of postmortem SCAD diagnosis and molecular autopsy encompass multiple dimensions: rigorous autopsy practice, specimen handling and documentation, prudent interpretation of genetic results, family communication and follow-up, as well as regulatory and organizational adjustments. The adoption of standardized diagnostic pathways, multidisciplinary teams, and policies for protecting privacy and counseling not only enable more accurate diagnoses, but also reduce legal risk, improve preventive care for family members, and transform forensic investigations into a public health opportunity.

## 8. Conclusions and Future Perspectives

SCAD is a potentially life-threatening condition, and its role as a cause of acute coronary syndrome and, in more severe cases, SCD is now well recognized, particularly in populations once considered low-risk, such as young women without significant atherosclerosis. What makes SCAD unique from atherosclerotic coronary heart disease is its pathophysiological substrate, in which arterial wall fragility and intramural hemorrhage, rather than lipid plaque rupture, represent the primary mechanism. This distinction has important implications for both clinical management and forensic recognition, as it requires high diagnostic attention in patient groups who would not otherwise be considered at high cardiovascular risk.

The integration of conventional autopsy targeted histological investigations, and MA today represents the gold standard for the recognition of this pathology. Macroscopic inspection of the coronary arteries, combined with histological confirmation of medial discontinuity and intramural hematoma, allows for precise postmortem diagnosis, while genetic analysis adds a further dimension, revealing hereditary predispositions. This integrated approach not only clarifies the cause of death but also provides information on the mechanisms of arterial wall fragility, such as fibromuscular dysplasia or connective tissue diseases, with significant consequences for surviving family members.

Therefore, raising awareness of SCAD not only among pathologists, but also among cardiologists, emergency physicians, gynecologists, and general practitioners is essential. The clinical presentation of SCAD often mimics that of classic myocardial infarction, but the therapeutic implications differ invasive interventions such as percutaneous coronary angioplasty may carry a higher risk of complications, while in stable cases a conservative approach is preferable. For this reason, a timely and accurate diagnosis during life can reduce iatrogenic damage, optimize treatment, and ultimately lower the risk of fatal outcomes. Educational campaigns, the inclusion of SCAD in cardiology training programs, and the dissemination of updated clinical guidelines are therefore fundamental steps to improve recognition both in the acute phase and during follow-up.

The establishment of national registries, the standardization of autopsy protocols, and the expansion of genetic screening programs represent practical tools for translating knowledge about this disease into effective preventive strategies. Registries enable the systematic collection of epidemiological, clinical, and genetic data, enabling large-scale analyses that can identify risk factors, refine diagnostic criteria, and guide public health interventions. Standardized autopsy protocols, in turn, reduce diagnostic variability between centers, ensuring that SCAD cases are not overlooked or misclassified as other forms of coronary artery disease. Genetic screening programs in families of affected individuals extend the benefits of postmortem recognition to prevention, allowing for the early identification of at-risk individuals and the implementation of lifestyle modifications, surveillance strategies, or prophylactic therapies.

Postmortem diagnosis is crucial, as recognizing SCAD in an apparently healthy heart not only provides a precise explanation of the cause of death but also opens a window into the future health of the entire family, transforming a tragic event into an opportunity for prevention for subsequent generations. This transformative potential underscores the broader value of pathology and forensic medicine as disciplines that connect case resolution to public health protection. When properly implemented, SCAD recognition at autopsy is not simply a retrospective exercise, but a proactive measure capable of influencing clinical practice, shaping guidelines, and guiding preventive strategies.

Furthermore, the medicolegal implications of accurate recognition of SCAD should not be underestimated. In cases of sudden unexplained death, especially in young people, families often seek definitive answers. An autopsy report that clearly identifies SCAD as the cause of death not only provides a sense of closure, but also has implications for liability, clinical review, and health policy. The integration of molecular autopsy into routine practice raises additional ethical considerations, including the management of genetic information, family counseling, and the balance between confidentiality and the duty of prevention.

Addressing these challenges requires multidisciplinary collaboration between pathologists, cardiologists, geneticists, ethicists, and forensic experts, ensuring that postmortem findings are responsibly integrated into both the clinical and public health contexts.

In conclusion, SCAD represents the evolution of the meeting point between cardiovascular pathology, clinical medicine and forensic practice. Its recognition as a cause of acute coronary syndrome and sudden death underscores the need for increased multidisciplinary awareness, robust diagnostic protocols, and systematic preventive strategies. By combining conventional and molecular approaches, promoting education and registry development, and thoughtfully addressing medicolegal and ethical dimensions, the medical community can transform the lessons learned from each SCAD case into concrete improvements in patient care and family health protection.

## Figures and Tables

**Figure 1 ijms-26-11072-f001:**
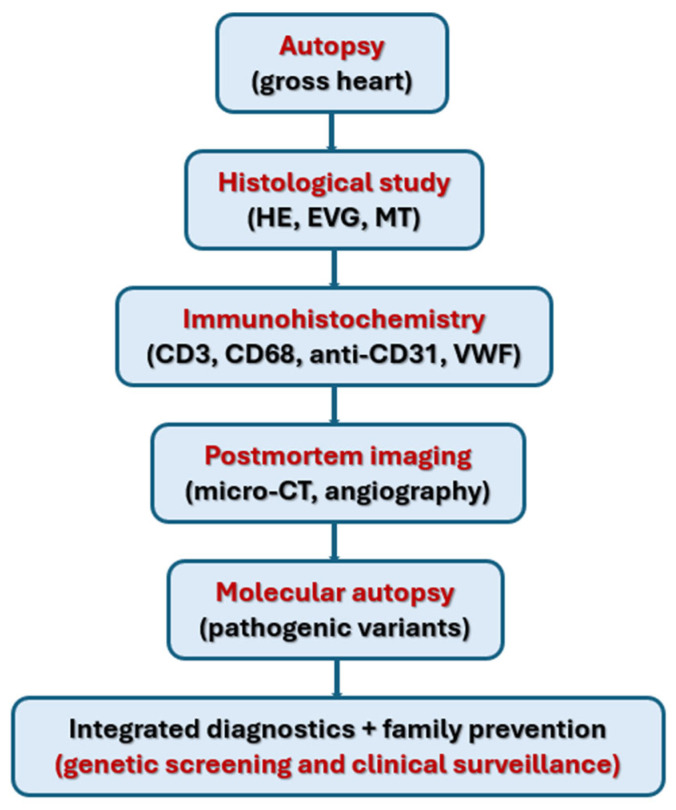
Integrated post-mortem diagnostic pathway.

**Table 1 ijms-26-11072-t001:** SCAD vs. coronary atherosclerosis.

	SCAD	Atherosclerosis
Etiology	Non-atherosclerotic, spontaneous dissection	Lipid deposition, chronic inflammation
Population affected	Young women, childbearing age, postpartum	Men >50 years, classic CV risk factors
Predisposing factors	FMD, connective tissue diseases, hormones, hemodynamic stress	Hyperlipidemia, hypertension, diabetes, cigarette smoking
Macroscopic examination	Intramural hematoma, false lumen	Atherosclerotic plaque with stenosis/rupture
Histological examination	Medial separation, elastic fiber degeneration	Fat accumulation, chronic inflammation, fibrosis
Clinical impact	SCD in apparently healthy hearts	Myocardial infarction, SCD, heart failure

Data summarized from references [1,3,4,5].

**Table 2 ijms-26-11072-t002:** Major genes associated with SCAD.

Major Genes	Frequency	Associated Condition	Mechanism
COL3A1	~20%	Vascular Ehlers–Danlos syndrome	Reduced arterial wall resistance
FBN1	Sporadic	Marfan syndrome	Connective tissue alterations, vascular dilations
TLN1	~14%	Family SCAD	Cell adhesion, vascular fragility
TSR1	~8%	SCAD	Vascular regulation
SMAD3	Isolated cases	Loeys–Dietz syndrome	Altered TGF-β pathway
FMR1 premutation	Rare	Unclear mechanisms	RNA/miRNA alterations

Data summarized from references [9,16,19,20,21].

**Table 3 ijms-26-11072-t003:** Pathological features of SCAD.

**Macroscopy**	- Intramural hematoma (most often between the media and adventitia, rarely between the intima and media). - Segmental thickening of the arterial wall. - Concentric or eccentric stenosis of the true lumen. - Variable in extent from mm to cm, sometimes multi-vessel. - Left anterior descending artery most affected. - In cross-section: false lumen with irregular reddish-purple/brown cavity compressing the true lumen. With intimal rupture: communicating orifice between the true and false lumens. Without intimal rupture (primary hematoma): smooth endoluminal surface, elongated stenosis.
**Histology**	- Separation of elastic and collagen fibers in the media. - Interposition of coagulated blood or serous fluid. - Distorted or interrupted internal elastic lamina. - Focal myxoid degeneration, fragmentation of elastic fibers. - Inflammatory infiltrate, generally mild, composed of T lymphocytes, macrophages, occasional eosinophils.
**Associated** **alterations**	- Fibromuscular dysplasia: medial disorganization, fibrous thickening, and smooth muscle cell rarefaction. - Connective tissue diseases: reduced fibril density and altered architecture of elastic and collagen fibers.
**Deadly** **complications**	- Acute myocardial ischemia: transmural or subendocardial coagulative necrosis. - Myocardial fiber swaying, interstitial edema, capillary congestion. - In very rapid deaths: loss of transverse striations, modest neutrophilic infiltrate.

Data summarized from references [16,22].

## Data Availability

No new data were created or analyzed in this study. Data sharing is not applicable to this article.

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
