# Peer review of "Unraveling Spontaneous Coronary Artery Dissection in Sudden Cardiac Death: Integrating Pathology, Genetics, and Molecular Autopsy"

_ijms, 2025, doi:10.3390/ijms262211072_

Round 1

Reviewer 1 Report

Comments and Suggestions for Authors

Although this paper appears to be structured as a narrative review, it is nevertheless essential to provide transparency regarding how the available evidence was identified, selected, and synthesized, in order to enhance both credibility and reproducibility. Therefore, the process of literature collection should be clearly described:

  1. Which databases were searched (e.g., PubMed, Scopus, Web of Science)
  2. The combinations of keywords and MeSH terms used
  3. The overall inclusion and exclusion criteria applied It should also be clarified how the articles were selected: did the review follow a predefined thematic framework, or did the synthesis emerge inductively from the findings and readings? Form a separate subsection

Lines 141–146 and 158–170 currently lack corresponding citations. Each claim within these segments should be supported by at least one reference. In addition, all tables included in the manuscript must clearly cite their data sources, either within the table captions or as footnotes, to ensure proper attribution and traceability of the evidence presented.

Author Response

Dear Reviewer1,

We sincerely thank the reviewer for his methodological suggestions, which we believe are extremely helpful in improving the transparency and reproducibility of our work.

Comments 1: Therefore, the process of literature collection should be clearly described:

1.Which databases were searched (e.g., PubMed, Scopus, Web of Science)

2.The combinations of keywords and MeSH terms used

3.The overall inclusion and exclusion criteria applied It should also be clarified how the articles were selected: did the review follow a predefined thematic framework, or did the synthesis emerge inductively from the findings and readings? Form a separate subsection

Response 1:We have therefore introduced a new subsection entitled “Materials and Methods” (positioned immediately after the Introduction and highlighted in red), in which we describe in detail the bibliographic search strategy, the inclusion/exclusion criteria and the approach adopted for the narrative synthesis.

Comments 2: Lines 141–146 and 158–170 currently lack corresponding citations. Each claim within these segments should be supported by at least one reference.

Response 2: Appropriate citations have been added to the indicated lines (highlighted in red) to support the missing statements.

Comments 3: In addition, all tables included in the manuscript must clearly cite their data sources, either within the table captions or as footnotes, to ensure proper attribution and traceability of the evidence presented.

Response 3: All tables (Table 1–3) have been updated with notes below the tables (highlighted in red) specifying the sources of the data and summaries reported, thus ensuring correct attribution and traceability of the information.

Thank you and kind regards,

The Authors

Reviewer 2 Report

Comments and Suggestions for Authors

This review is dedicated to spontaneous coronary artery dissection (SCAD) as an important reason of non-atherosclerotic cause of acute coronary syndrome. It is characterized by the development of a false lumen within the coronary arterial wall and appropriate narrowing or occlusion of the true lumen. This condition is  poorly recognized and accounts for a substantial proportion of sudden cardiac death among young and very often healthy women. Authors described pathogenic mechanisms and predisposing conditions that led to SCAD as well  autopsy findings including histochemical and immunohistochemical analyses. They also discuss the role of molecular autopsy as a diagnostic and preventive tool.

The paper is well written and illustrated. I have no any concerns. This review sounds good and deserves to be published.

Author Response

Dear Reviewer 2,

We sincerely thank the reviewer for his careful reading of our manuscript and for his kind words of appreciation. We are pleased that the Reviewer recognized the clarity of the discussion and the illustrative value of the work. Since there were no further comments or requests for amendments, no substantial changes were made to the text.

We thank you again for your positive opinion and for the time spent reviewing it.

Kind regards,

The Authors

Reviewer 3 Report

Comments and Suggestions for Authors

The authors of this review article agree that the problem of Spontaneous coronary artery dissection (SCAD), although not as common in cardiology practice, does exist. Sudden cardiac death in relatively young individuals is a constant problem. And very often, the true cause of death cannot be determined even after a forensic autopsy. This causes frustration among both medical professionals and the relatives of the deceased, because the default diagnosis of acute heart failure does not reveal the true cause of death.

This medical review provides a descriptive analysis of the external macroscopic manifestations of SCAD, microscopic examinations, and the new scientific field of molecular autopsy (MA). This is postmortem genetic analysis, which is performed when the cause of death remains unknown after a traditional autopsy. It helps identify genetic mutations that could have led to sudden death. Traditional molecular autopsy focuses on direct DNA sequencing of protein-coding exons of four genes, including the three major LQTS genes (KCNQ1, KCNH2, SCN5A) and the CPVT gene (RYR2).

This research has become possible thanks to the development and implementation of new, modern technologies.

Author Response

Dear Reviewer3,

We sincerely thank the reviewer for his careful reading and extremely pertinent comments.

We fully agree with the observation regarding the importance of SCAD as an underdiagnosed cause of sudden cardiac death in young, clinically healthy subjects, as well as the frustration arising from cases in which the postmortem diagnosis remains indeterminate.

We are grateful that the reviewer appreciated the section of the manuscript dedicated to molecular autopsy, which represents one of the most innovative aspects of modern forensic medicine. Indeed, as correctly emphasized, molecular autopsy allows us to identify genetic mutations potentially responsible for sudden death even in the absence of obvious macroscopic or histological alterations.

In our work, we have expanded this perspective, including not only the classic genes associated with inherited arrhythmia syndromes (KCNQ1, KCNH2, SCN5A, RYR2), but also genes implicated in vascular fragility and predisposition to SCAD, such as COL3A1, FBN1, TLN1, SMAD3, and FMR1, according to the most recent literature evidence.

We have also added a short sentence to the text to clarify this aspect, highlighting the contribution of next-generation sequencing (NGS) technologies in improving the diagnostic yield of molecular autopsy (section 5, opening paragraph, highlighted in yellow).

We again thank the reviewer for his positive and constructive comments, which fully enhance the multidisciplinary approach of our work.

Best regards,

The Authors

Round 2

Reviewer 1 Report

Comments and Suggestions for Authors

Thank you for making the changes according to previous suggestions. Good luck!